# Chemical Composition of *Salix koreensis* Anderss Flower Absolute and Its Skin Wound Healing Activities In Vitro

**DOI:** 10.3390/plants11030246

**Published:** 2022-01-18

**Authors:** Nan Young Kim, Kyung Jong Won, Ha Bin Kim, Do Yoon Kim, Mi Jung Kim, Yu Rim Won, Hwan Myung Lee

**Affiliations:** 1Division of Cosmetic and Biotechnology, College of Life and Health Sciences, Hoseo University, Asan 31499, Korea; rlasksdud01@naver.com (N.Y.K.); rlagkqls24@naver.com (H.B.K.); doyoon@hoseo.edu (D.Y.K.); k7716708@naver.com (M.J.K.); bb_bb22@naver.com (Y.R.W.); 2Department of Physiology and Medical Science, School of Medicine, Konkuk University, Seoul 05029, Korea; kjwon@kku.ac.kr

**Keywords:** *Salix koreensis* Anderss, absolute, skin wound healing, keratinocytes, proliferation, migration, collagen synthesis

## Abstract

*Salix koreensis* Anderss (SKA) has been used traditionally to treat inflammation, pain, and edema. SKA has anti-inflammatory and antioxidant effects, but no study has examined its effects on skin wound healing. Here, we aimed to investigate the effects of the absolute extracted from SKA flower (SKAFAb) on skin wound healing-associated responses in keratinocytes. SKAFAb was produced using a solvent extraction method and its chemical composition was analyzed by gas chromatography/mass spectrometry. The effects of SKAFAb on HaCaT cells (a human epidermal keratinocyte cell line) were investigated using a Boyden chamber and 5-bromo-2′-deoxyuridine incorporation, sprout outgrowth, immunoblotting, enzyme-linked immunosorbent, and water-soluble tetrazolium salt assays. Sixteen constituents were identified in SKAFAb. SKAFAb promoted HaCaT cell proliferation, migration, and type I and IV collagen productions. SKAFAb increased sprout outgrowth and increased the phosphorylations of serine/threonine-specific protein kinase (Akt), c-Jun NH2-terminal kinase, extracellular signal-regulated kinase1/2, and p38 mitogen-activated protein kinase (MAPK) in HaCaT cells. These results indicate that SKAFAb promotes keratinocyte proliferation and migration, probably by activating Akt and MAPK signaling pathways, and induces collagen synthesis in keratinocytes. SKAFAb may be a promising material for promoting skin wound healing.

## 1. Introduction

The skin protects the body against harmful exogenous events and can rapidly and effectively restore the integrities of damaged tissues [1]. Skin wound healing immediately after injury proceeds in four successive but overlapping stages, namely, hemostasis, inflammation, proliferation, and remodeling [2]. The regulations of these processes are associated with complex interactions between cells, chemokines, cytokines, growth factors, extracellular matrix (ECM) components, and blood elements [3]. Wound healing can be disrupted by factors, such as infection, tissue necrosis, stress, and disease [2]. An abnormality in any one of these stages of healing can disrupt the healing process [1]. Such disruptions can cause undue stress and place financial burdens on patients and their families. Therefore, appropriate and effective pharmacotherapy, management, and treatments are required to achieve complete recovery of damaged skin.

During wound healing, keratinocytes (the predominant cells in the epidermis) act as a crucial player in the re-epithelialization process [2]. Re-epithelialization is essential and triggered to restore the epidermis, which is the most important contributor to skin barrier function [4]. In damaged skin, epidermal keratinocytes exhibit re-epithelialization-promoting biological activities, such as migration, proliferation, and differentiation [5,6]. The migratory and proliferative activities of keratinocytes are stimulated by various factors, such as growth factors and cytokines, during the proliferative stage of skin wound healing [7]. These factors are known to be activated by signals mediated by intricate intracellular signaling cascades, which include the serine/threonine-specific protein kinase (Akt) and mitogen-activated protein kinase (MAPK) pathways [8,9]. Collagen has also been reported to be required for healing-related responses in the skin during all healing phases [10,11]. For example, collagen plays a pivotal role in attracting cells, such as keratinocytes and fibroblasts to wounds [11], and products resulting from its cleavage have been reported to participate in the enhanced migrations and proliferation exhibited by wound healing-related cells [12]. Therefore, it would appear that suitable regulation of the biological activities in keratinocytes might provide a strategy to promote skin repair.

Many therapies have been devised to promote skin repair, but their efficacies are limited [13]. On the other hand, it has been reported that natural products can enhance skin repair [14], and in particular, plants continue to be used in traditional medicine for this purpose [15,16]. Many plants or plant extracts have been reported to have broad therapeutic abilities and minimal adverse effects [15,16]. Thus, plants and their extracts have become a focus for the development of effective, safe alternative therapies for the treatment of skin wounds [17]. Many plants of the Salix genus have been used traditionally to prevent oral inflammation and dental caries and to treat various conditions, such as fever, pain, and inflammation [18]. Furthermore, they have also been reported to have antioxidant, anti-obesity, hepatoprotective, and anti-cancer activities [18].

*Salix koreensis* Anderss (SKA; family Salicaceae), also reported as the Korean willow is a deciduous broad-leaved tree that grows in wetlands and is well distributed in East Asia, including Korea and China [19]. SKA has been reported to have antioxidant, anti-inflammatory, and hepatoprotective effects [20,21]. However, the effects of SKA on skin wound healing have not been investigated. In addition, the biological activities of SKA flowers or their extracts are unknown. Thus, in this study, we aimed to investigate the effects of the absolute extracted from SKA flower (SKAFAb) on wound healing-linked responses in HaCaT cells (a human keratinocyte cell-line).

## 2. Results

### 2.1. Chemical Composition of SKAFAb

Gas chromatography/mass spectrometry (GC/MS) identified 16 constituents in SKAFAb (Table 1, Figure 1). Of these 16 components, the component with the largest peak area was 9-tricosene, (Z)- (26.69%; peak 10), followed by docosane (23.36%; peak 9), tricosane (14.26%; peak 8), 2-nonadecanone (10.16%; peak 5), verbenone (4.38%; peak 2), eicosane (4.31%; peak 7), pentacosane (3.81%; peak 11), and 1-eicosanol (3.24%; peak 14) (Table 1 and Figure 1).

### 2.2. Proliferative and Migratory Activities of Keratinocytes Exposed to SKAFAb

Keratinocyte proliferation and migration are essential responses to skin injury [22,23]. To determine the proliferative and migratory activities of keratinocytes exposed to SKAFAb, we first tested whether SKAFAb (1–250 μg/mL) affected HaCaT cell viability using a water-soluble tetrazolium salt (WST) assay. SKAFAb significantly increased HaCaT cell viability at concentrations of 50 to 250 μg/mL, but had no effect at 1 or 10 μg/mL (Figure 2a). Next, we evaluated the effect of SKAFAb (1–250 μg/mL) on proliferative activity using the 5-bromo-2′-deoxyuridine (BrdU) incorporation assay. SKAFAb significant increased HaCaT cell at 10 to 200 μg/mL, and this peaked at 100 μg/mL (149.26 ± 8.51% vs. untreated controls) (Figure 2b).

The effect of SKAFAb on migratory activity was analyzed at SKAFAb (1–250 μg/mL) using the Boyden microchemotaxis chamber assay. At concentrations of 50 to 150 μg/mL, SKAFAb showed a significant increase in HaCaT cell migration (Figure 3a,b), and this was at a maximum at 100 μg/mL (165.87 ± 5.16% vs. untreated controls) (Figure 3b).

### 2.3. Effect of SKAFAb on Keratinocyte Sprout Outgrowth

The proliferative and migratory effects of SKAFAb on HaCaT cells were assessed using a collagen sprout outgrowth assay, which is widely used to simultaneously assess cell proliferation and migration in vitro [24,25]. SKAFAb (1–250 μg/mL) significantly increased sprout outgrowth by HaCaT cells at 50 to 150 μg/mL (Figure 4a,b), and produced maximum outgrowth at 100 μg/mL (158.15 ± 8.84% vs. untreated controls) (Figure 4b).

### 2.4. Effects of SKAFAb on Signaling Kinases in Keratinocytes

Akt and MAPKs are important signaling molecules and are associated with the migratory and proliferative activities of HaCaT cells [26]. Thus, we investigated the possible involvements of these kinases in migration and proliferation in SKAFAb (1–250 μg/mL) treated HaCaT cells by Western blotting. SKAFAb significantly increased p38 MAPK phosphorylation at 10 to 250 μg/mL (Figure 5a,b), extracellular signal-regulated kinase1/2 (ERK1/2) phosphorylation at 100 to 200 μg/mL (Figure 5a,c), c-Jun NH2-terminal kinase (JNK) phosphorylation at 10 to 250 μg/mL (Figure 5a,d), and Akt phosphorylation at 50 to 250 μg/mL (Figure 5a,e). Maximum phosphorylation levels of these kinases were observed at a SKAFAb concentration of 100 μg/mL for p38 MAPK (439.94 ± 42.62% vs. untreated controls) and at a concentration of 150 μg/mL for ERK1/2, JNK, and Akt (326.62 ± 35.27%, 385.83 ± 3.21%, and 264.53 ± 22.39%, respectively, vs. untreated controls) (Figure 5b–e).

### 2.5. Collagen Synthesis in Keratinocytes Exposed to SKAFAb

Collagen plays important role in the proliferation, attachment, and migration of cells, which are all associated with skin wound healing [10,27]. Thus, we examined the effects of SKAFAb on collagen synthesis in keratinocytes using a sandwich enzyme-linked immunosorbent assay (ELISA). As shown in Figure 6, In conditioned media of HaCaT cells, SKAFAb (1–150 μg/mL) increased type I and IV collagen levels in a concentration-dependent manner. Increased type I collagen level was significant at 150 μg/mL, whereas type IV collagen level was significant at SKAFAb concentrations of 50 and 150 μg/mL (from 50 to 150 μg/mL). The levels of both collagens peaked at a concentration of 150 μg/mL (127.33 ± 5.72% vs. untreated controls for type I collagen, Figure 6a; and 136.54 ± 3.73% for type IV collagen, Figure 6b).

## 3. Discussion

The skin healing effects of plants and plant extracts have been extensively investigated with a view toward developing agents that promote wound healing agents because of their minimal side effects and low development costs [28,29]. In the present study, we found that SKAFAb induced the proliferation and migration of HaCaT cells and increased HaCaT cell sprout growth in a collagen sprouting assay, which can simultaneously observe migration and proliferative ability [24,25,30]. These results confirm that SKAFAb can stimulate the proliferation and migration of keratinocytes. However, these observations have not been previously reported. To the best of our knowledge, our results provide the first evidence that SKAFAb promotes keratinocyte migration and proliferation. It has been well established that keratinocyte migration and proliferation are essentially required for the re-epithelialization of wounded skin [8]. Re-epithelialization is an important process for restoring the new epidermis for the healing of damaged skin [2,4]. Therefore, our findings suggest that SKAFAb might positively aid skin wound healing.

In the present study, GC/MS analysis of SKAFAb identified 16 components that might be responsible for the ability of SKAFAb to increase HaCaT cell migration and proliferation. Several of the compounds identified were associated with migratory or proliferative activity against various types of cells. For example, a literature search revealed that one of these components, nonanal, enhanced proliferative and migratory activities and upregulated growth factor genes, such as keratinocyte growth factor in human hair follicle dermal papilla cells [31]. Verbenone had an antiproliferative effect on ovarian carcinoma cells [32]. Moreover, Figueiredo et al. reported that 1-eiocanol and 2-nonadecanone were the main constituents of the *Kielmeyera coriacea* extract inhibiting migration of murine melanoma cells and suggested that they might be components responsible for the ability of *Kielmeyera coriacea extract* to inhibit the migration of melanoma cells [33]. These reports imply that SKAFAb may affect keratinocyte migration and/or proliferation. In the present study, we confirmed the ability of SKAFAb to promote HaCaT cell migration. However, none of the components detected has been reported to affect keratinocyte migratory or proliferative activities or other wound healing-linked responses. Therefore, we suggest that further studies be conducted to investigate the effects of these compounds on keratinocytes.

Wound healing is regulated by intracellular signaling pathways, and MAPKs are known to be associated with the regulations of many cellular events, including cellular proliferation and migration [3]. The MAPK family is composed of three major subfamilies, namely, p38 MAPK, ERK1/2, and JNK, which have all been reported to mediate keratinocyte migration and proliferation [26,34]. Activation of p38 MAPK signaling has been reported to be associated with increased keratinocyte proliferation and migration [35,36], and its inhibition was reported to inhibit keratinocyte migration and proliferation [37]. Inhibition of ERK1/2 phosphorylation also suppressed keratinocyte migration and proliferation [38,39], whereas its upregulation increased both [26,40,41]. Zhao et al. reported that JNK activation enhanced keratinocytes migration and proliferation [26], whereas others have reported JNK activation had no effect [35,42]. These reports imply that p38 MAPK and ERK1/2 signaling molecules may act as important contributors to the migratory and proliferative activity of keratinocytes, but JNK may not. In the present study, SKAFAb at 10 to 250 μg/mL enhanced the phosphorylation levels of p38 MAPK and JNK and at 100 to 200 μg/mL enhanced ERK1/2 phosphorylation in HaCaT cells. Thus, our observations indicate that SKAFAb enhances the migration and proliferation of HaCaT cells by phosphorylating MAPKs. In addition, Akt activation has also been reported to mediate keratinocyte migration and proliferation [40,41,43]. We found that SKAFAb at concentrations of 50 to 250 μg/mL enhanced Akt phosphorylation in HaCaT cells, which indicates that SKAFAb may promote migration and proliferation through Akt activation. Therefore, SKAFAb might have mediated migratory and proliferative responses by activating Akt and/or MAPK signal transduction.

Collagen is a major component of ECM in the dermis and is known to be synthesized and secreted by keratinocytes or fibroblasts and to function as a scaffold for keratinocytes [44]. Collagen is importantly related to skin wound healing and to have crucial functions during the proliferative and remodeling phases [10,44]. During the proliferative phase of the wound healing process, collagen facilitates keratinocyte migration across wounds [8,44], and its cleavage products can also promote migratory and proliferative responses in keratinocytes [12]. Interstitial collagen type I is the primary collagen of skin and a key player in skin healing [44,45,46,47], whereas basement collagen type IV is involved in membrane formation [44,48]. Woodley et al. reported that type I and type IV collagens increase keratinocyte migration [49]. Previous studies have shown many plant extracts enhance the syntheses and secretions of type I and/or IV collagens by HaCaT cells [25,30,50], which implies keratinocytes can synthesize and secrete collagen types I and IV. In the present study, SKAFAb increased collagen type I at concentrations of 150 μg/mL and 50 and IV syntheses 150 μg/mL in HaCaT cells. Therefore, our findings suggest that SKAFAb may have a positive effect on skin healing by promoting the syntheses of collagen types I and IV by keratinocytes.

## 4. Materials and Methods

### 4.1. Materials

Trypsin-ethylenediamine tetra-acetic acid (EDTA) and fetal bovine serum (FBS) were purchased from Gibco BRL (Gaithersburg, MD, USA). Phosphate buffered saline (PBS) and Dulbecco’s modified eagle medium (DMEM) were obtained from Welgene (Daegu, Korea). Penicillin/streptomycin (P/S) was purchased from Hyclone (Logan, UT, USA), and recombinant human (rh) EGF from R&D Systems (Minneapolis, MN, USA). Dimethyl sulfoxide (DMSO) and bovine serum albumin (BSA) were from Merck Sigma (St. Louis, MO, USA). Type I collagen was obtained from BD Bioscience (Franklin Lakes, NJ, USA). The EZ-CyTox kit was from DAEIL LAB Service (Seoul, Korea). The antibodies used were as follows; anti-JNK, anti-phospho JNK, anti-p38 MAPK, anti-phospho p38 MAPK, anti-ERK1/2, anti-phospho ERK 1/2, anti-Akt, anti-phospho Akt, anti-rabbit immunoglobulin G (IgG), anti-mouse IgG (Cell Signaling, Beverly, MA, USA), and polyclonal anti-type I and IV collagens and monoclonal anti-type I and IV collagens (Abcam, Cambridge, UK). β-Actin was purchased from Merk Sigma.

### 4.2. Extraction of Salix koreensis Anderss Flower Absolute

SKA flowers were collected trees near the farm of the Division of Cosmetic Science, Hoseo University, Baebang-eup, Asan, Korea (36°74′05.9″ N 127°07′80.7″ E) and identified by Jong-Cheol Yang (Division of Forest Biodiversity and Herbarium, Korea National Arboretum, Pocheon, Korea). A voucher specimen (NO. SK-0003) was kept at the Herbarium of the College of Life and Health Science (Hoseo University, Korea). Absolute was extracted from SKA flowers by solvent extraction, as in previous reports [30]. In brief, SKA flowers (2.05 kg) were completely immersed in n-hexane (Samchun, Pyeongtaek, Korea) at room temperature (RT) for 1 h. Extracts were obtained, and the solvent was removed in a rotary evaporator (EYELA, Tokyo, Japan) at 25 °C under vacuum. The dark yellow waxy residue (concrete) obtained was mixed with ethanol (Samchun) and left at −20 °C for 12 h, filtered through a sintered funnel, and then the ethanol was removed by evaporation at 35 °C to obtain a light-yellow anhydrous wax (SKAFAb; 2.65 g, yield 0.13% *w/w*), which was stored at −80 °C).

### 4.3. Identification of Components in SKAFAb

Components of SKAFAb were analyzed by GC/MS at the Korean Basic Science Institute (KSBI, Seoul, Korea) as in a previous report [30]. GC/MS analysis was conducted using a 7890BGC/7010QQQ MS instrument (Agilent, Palo Alto, CA, USA) and a DB5-MS capillary column (30 m × 0.25 mm, film thickness 0.25 μm). Helium was used as a carrier gas and its flow rate was 1 mL/min. The injector port, ion source, and interface temperatures were 280, 300, and 300 °C, respectively. The GC oven was set up as follows; 40 °C for 2 min, 40 to 230 °C at 2 °C/min, 230 to 300 °C at 5 °C/min, and 300 °C for 5 min. The split ratio was 1:10. Masses were scanned from m/z 50 to 800. RIs were determined using the Kovats method using standard C_7_-C_40_ n-alkanes, and compounds were identified by comparing their RI values with Kovats indices [51] and by matching their MS fragmentation patterns with those in the Wiley7NIST0.5L Mass Spectral library and catalogs. GC/MS data were obtained as analyzed by Adams [52].

### 4.4. Cell Culture

HaCaT cells (a human keratinocyte cell line) were obtained from the Daegu Gyeongbuk Institute for Oriental Medicine Industry (Gyeongsan City, Korea) and cultured in DMEM with 10% FBS and 1% P/S. Cells were incubated in a humidified 95% air/5% CO_2_ atmosphere at 37 °C, and for experiments were cultured until 70–80% confluent.

### 4.5. Cell Viability Assay

Cell viability analysis was performed by a WST assay using the EZ-CyTox kit. HaCaT cells were seeded at 5 × 10^3^ cells/well into 96-well microtiter plates. Cells were incubated with different concentrations of SKAFAb (dissolved and diluted in DMEM containing 0.5% DMSO) for 36 h and then incubated with EZ-CyTox reagent (10 μL/well) for 30 min at 37 °C. Cell viabilities were measured using a multi-well plate reader (Synergy 2, Bio-Tek Instruments, Winooski, VT, USA) at 450 nm.

### 4.6. Proliferation Assay

Cell proliferation was assessed using a BrdU incorporation assay (Roche, Indianapolis, Indiana, USA), as in a previous report [24]. In brief, HaCaT cells were plated in a 96-well plate at 3 × 10^3^ cells/well, treated with various concentrations of SKAFAb (dissolved and diluted in DMEM containing 0.5% DMSO) or rhEGF (50 ng/mL) for 36 h, and then incubated with BrdU-labeling solution (10 μM) for 12 h at 37 °C. After denaturing DNA, cells were incubated with peroxidase labeled anti-BrdU monoclonal antibody at RT for 90 min. The BrdU antibody complexes formed were detected using a luminometer (Synergy 2, Bio-Tek Instruments).

### 4.7. Migration Assay

Cell migration was assessed using a 48-well Boyden microchemotaxis chamber (Neuro Probe Inc., Gaithersburg, MD, USA), as in a previous report [30]. Briefly, lower chambers were loaded with DMEM containing 0.1% BSA and different concentrations of SKAFAb or rhEGF (5 ng/mL). A membrane coated with type Ι collagen was then laid over medium in lower chambers. Upper chambers were loaded with cells (5 × 10^4^ cells/well) in DMEM containing 0.1% BSA. Chambers were then incubated for 210 min at 37 °C. Membranes were removed and then fixed and stained using Diff-Quick (Baxter Healthcare, Miami, FL, USA). The lower part images of the membrane were captured in three randomly selected regions of each well using an optical microscope (×200). The number of migrated cells in captured images of each well was counted and the mean values were calculated. Each experiment was repeated four times. Migration level was expressed as a percentage of control (the untreated state).

### 4.8. Collagen Sprout Assay

Simultaneous assessments of HaCaT cell proliferation and migration were performed using a collagen sprout assay, as previously reported [30]. Briefly, cells (2.5 × 10^7^ cells/mL) were mixed with type I collagen, 10× DMEM, and 1 N NaOH (pH 7.2), and spotted in the wells of a 24-well cell culture plate. After drying for 20 min, spots were incubated with or without SKAFAb or rhEGF (50 ng/mL) for 48 h at 37 °C in a CO_2_ incubator. Spots and sprouts were fixed and stained using Diff-Quik solution. The images of cell spots were captured in three randomly selected regions of each well using an optical microscope at 100-fold magnification. The sprout outgrowths in the images were determined by measuring the average length from the cell spot edge to the cell sprout edge in three chosen parts (longest, medium, and shortest length) using Image J software (Version Java 1.8.0., NIH, Bethesda, MD, USA). The mean values of sprout outgrowth length in each well were calculated. Each experiment was repeated four times. Sprout outgrowth length was expressed as a percentage of control (the untreated state).

### 4.9. Collagen Synthesis Assay

Collagen synthesis was assessed using an ELISA, as previously described [30]. Briefly, HaCaT cells (5 × 10^5^ cells/dish) were seeded in 100-mm culture dishes and incubated with different concentrations of SKAFAb for 48 h at 37 °C. Collected media were centrifuged sequentially at 500, 800, and 1000× *g* for 10 min. The supernatants (conditioned media; 100 μL/well) were collected and added to 96-well microtiter plates coated with capture antibody (type I or IV collagen monoclonal antibody), followed by incubation with biotin-conjugated collagen type I or IV polyclonal antibody (dilution 1:2000 in 1% BSA/PBS) for 90 min at RT. After washing wells with PBS, each well was treated with streptavidin-horseradish peroxidase conjugate (Roche) (diluted 1:5000 in 1% BSA/PBS) for 1 h at RT, washed with PBS, and treated with ECL solution (Thermo Fisher Scientific, Waltham, MA, USA). Luminescence was measured using a luminometer (Synergy 2, Bio-Tek Instruments).

### 4.10. Western Blotting

HaCaT cells were lysed using RIPA (radioimmunoprecipitation assay) buffer (Cell Signaling), and centrifuged at 17,000× *g* for 15 min at 4 °C. Supernatants were collected and protein concentrations were determined using DC protein assay reagents (Bio-Rad Laboratories, CA, USA). Proteins (50–130 μg/lane) were separated by 10% sodium dodecyl sulfate-polyacrylamide gel electrophoresis and transferred to polyvinylidene fluoride membranes (Millipore, St. Louis, MO, USA) at 4 °C. Membranes were blocked in 3% skim milk at RT for 2 h, washed with PBS containing 0.05% Tween-20, incubated with each target primary antibody (1: 1000–10,000 dilution), and then with horseradish peroxidase-conjugated secondary antibody at RT for 1 h. Protein bands were visualized using a chemiluminescence substrate and detected using a chemiluminescence imaging system (LuminoGraph, ATTO, Tokyo, Japan).

### 4.11. Statistical Analysis

Results are presented as means ± SEMs of the indicated number of experiments. The statistical significances of differences between two groups were analyzed using the Student’s t-test, and multiple comparisons were conducted by one-way ANOVA followed by a Tukey’s post hoc test in GraphPad Prism (version 5.0; Graphpad Software, Inc., San Diego, CA, USA). *p* values < 0.05 were considered to indicate significant differences.

## 5. Conclusions

In this study, 16 compounds were identified in SKAFAb. SKAFAb was found to induce the proliferation, migration, and sprout outgrowth of HaCaT cells. SKAFAb also enhanced the phosphorylations of p38 MAPK, ERK1/2, JNK, and Akt and increased the syntheses of type I and IV collagens in HaCaT cells. These results indicate that SKAFAb might promote skin wound healing by positively stimulating keratinocyte migratory and proliferative responses, possibly through MAPK and/or Akt-mediated signaling pathways, and by facilitating collagen synthesis by keratinocytes. Therefore, SKAFAb may offer a potential starting point for the development of therapeutic agents that accelerate skin repair.

## Figures and Tables

**Figure 1 plants-11-00246-f001:**
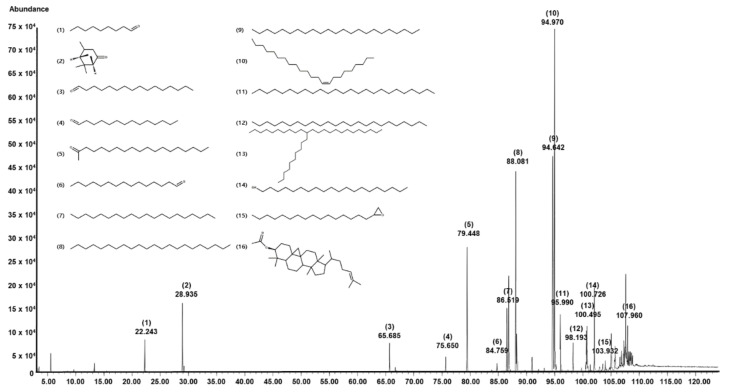
GC/MS total ion chromatogram of *Salix koreensis* Anderss flower absolute. Bracketed and unbracketed numbers at each peak express component numbers and retention times, respectively, of the 16 identified components (Table 1). Bracketed numbers and chemical structures on the upper left side of the chromatogram indicate the 16 identified compounds.

**Figure 2 plants-11-00246-f002:**
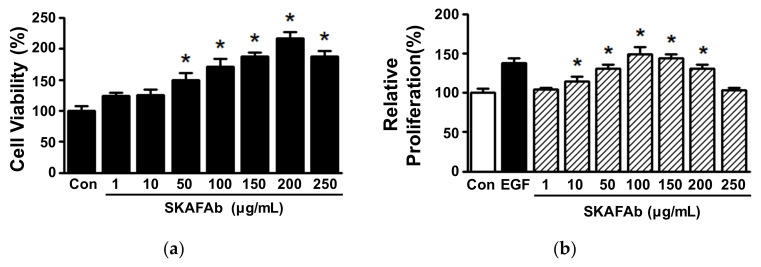
Effects of *Salix koreensis* Anderss flower absolute on keratinocyte viability and proliferation. (**a**) HaCaT cells were incubated in the presence or absence of *Salix koreensis* Anderss flower absolute (SKAFAb; 1–250 µg/mL) for 36 h. Cell viabilities were analyzed using a WST assay (*n* = 5). (**b**) HaCaT cells were treated with SKAFAb (1–250 µg/mL) for 36 h. Cell proliferations were assessed using a BrdU incorporation assay (*n* = 5). Recombinant human epidermal growth factor (rhEGF: 50 ng/mL) was used as a positive control. Percentages are of levels versus untreated cells (Con). Results are presented as means ± standard errors of means (SEMs). *p*-values by one-way analysis of variance (ANOVA) are <0.0001 for panel a and panel b, respectively. * *p* < 0.05 vs. untreated cells (by a Tukey’s post hoc test).

**Figure 3 plants-11-00246-f003:**
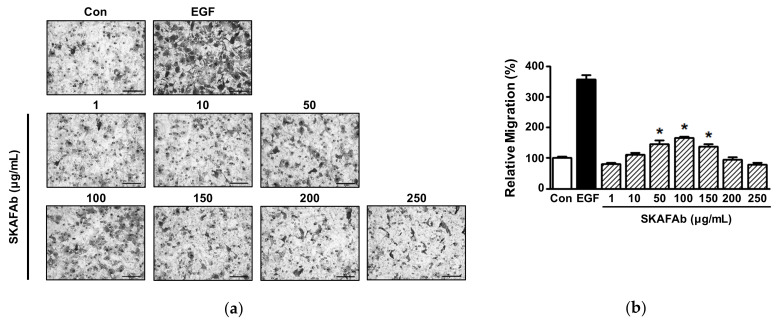
Effects of *Salix koreensis* Anderss flower absolute on keratinocyte migration. HaCaT cells were treated with or without *Salix koreensis* Anderss flower absolute (SKAFAb; 1–250 µg/mL) for 210 min. Cell migration levels were assessed using a Boyden microchemotaxis chamber assay. (**a**) Representative images. Black spots indicate migrating cells. (**b**) Statistical graph. Recombinant human epidermal growth factor (rhEGF: 5 ng/mL) was used as a positive control. Percentages are of levels versus untreated cells (Con). Results are presented as means ± SEMs (*n* = 4). Scale bar = 100 μm. *p*-values by one-way ANOVA are <0.0001 (**b**). * *p* < 0.05 vs. untreated cells (by a Tukey’s post hoc test).

**Figure 4 plants-11-00246-f004:**
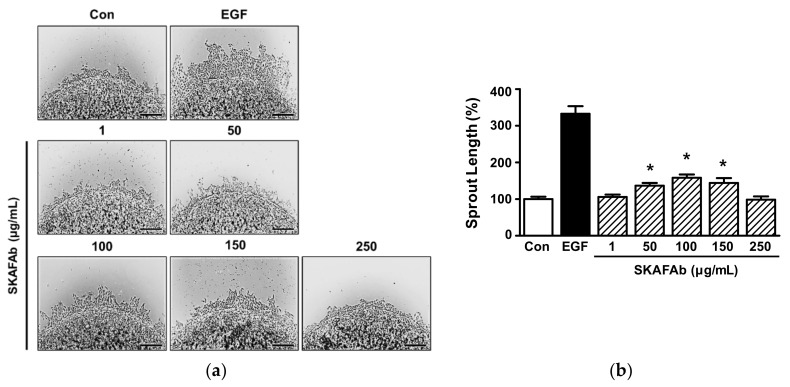
Effect of *Salix koreensis* Anderss flower absolute on keratinocyte sprout formation. (**a**) HaCaT cells mixed with collagen were spotted on a 24-well plate and incubated with or without *Salix koreensis* Anderss flower absolute (SKAFAb; 1–250 µg/mL) for 48 h. Cells were fixed and stained with Diff-Quick solution and photomicrographs were taken. Recombinant human epidermal growth factor (rhEGF: 50 ng/mL) was used as a positive control. Sprout lengths were measured using Image J. Scale bar = 100 μm. (**b**) Statistical graph. Percentages are of levels versus untreated cells (Con). Results are presented as means ± SEMs (*n* = 4). *p*-values by one-way ANOVA are <0.0001. * *p* < 0.05 vs. untreated cells (by a Tukey’s post hoc test).

**Figure 5 plants-11-00246-f005:**
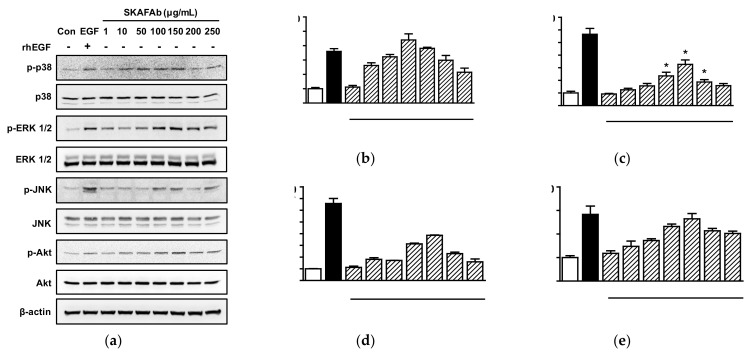
Kinase phosphorylations in keratinocytes exposed to *Salix koreensis* Anderss flower absolute. HaCaT cells were incubated with or without *Salix koreensis* Anderss flower absolute (SKAFAb; 1–250 µg/mL) for 10 min. Cell lysates were Western-blotted with kinase or β-actin antibodies. (**a**) Representative kinase expression images. (**b**–**e**) Plots of phosphorylated p38 MAPK (**b**), ERK1/2 (**c**), JNK (**d**), and Akt (**e**) expression levels versus SKAFAb concentration. The band intensity of phosphorylated protein was normalized to that of the corresponding non-phosphorylated protein. The percentage of the phosphorylated protein to non-phosphorylated protein under untreated state (Con) was considered as 100%. Recombinant human epidermal growth factor (rhEGF: 5 ng/mL) was used as a positive control. Results are presented as means ± SEMs (*n* = 3 for each protein). *p*-values by one-way ANOVA are <0.0001 for p-p38 (**b**), p-ERK1/2 (**c**), p-JNK (d), and p-Akt (**e**), respectively. * *p* < 0.05 vs. untreated cells (by a Tukey’s post hoc test). p-p38, phosphorylated p38 MAPK; p-ERK1/2, phosphorylated ERK 1/2; p-JNK, phosphorylated JNK; p-Akt, phosphorylated Akt.

**Figure 6 plants-11-00246-f006:**
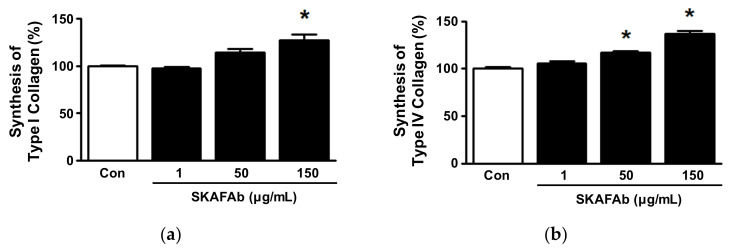
Syntheses of type I and IV collagens by keratinocytes exposed to *Salix koreensis* Anderss flower absolute. HaCaT cells were treated with or without *Salix koreensis* Anderss flower absolute (SKAFAb; 1–150 µg/mL) for 48 h. Conditioned media were subjected to sandwich ELISA using anti-type I (*n* = 3; (**a**)) or anti-type IV collagen antibody (*n* = 3; (**b**)). Collagen levels are expressed as percentages of levels in the conditioned media of untreated cells (Con). Values are presented as means ± SEMs. *p*-values by one-way ANOVA are 0.0126 (**a**) and 0.0015 (**b**), respectively. * *p* < 0.05 vs. untreated cells (by a Tukey’ post hoc test).

**Table 1 plants-11-00246-t001:** Chemical composition of *Salix koreensis* Anderss flower absolute.

No	Component Name	RT ^1^	RI ^2^		Area (%)	CAS No.
			Observed	Literature		
1	Nonanal	22.24	1104	1104	1.84	124-19-6
2	Verbenone	28.94	1201	1204	4.38	1196-01-6
3	Hexadecanal	65.69	1814	1814	1.78	629-80-1
4	Tetradecanal	75.65	2018	1940	0.90	124-25-4
5	2-Nonadecanone	79.45	2101	2101	10.16	629-66-3
6	Pentadecanal	84.76	2222	-	0.56	2765-11-9
7	Eicosane	86.52	2263	2000	4.31	112-95-8
8	Tricosane	88.08	2300	2300	14.26	638-67-5
9	Docosane	94.64	2465	2200	23.36	629-97-0
10	9-Tricosene, (Z)-	94.97	2474	2272	26.69	27519-02-4
11	Pentacosane	95.99	2500	2500	3.81	629-99-2
12	Tetracosane	98.19	2574	2400	1.70	646-31-1
13	Tetracosane, 11-decyl-	100.50	2665	-	0.54	55429-84-0
14	1-Eicosanol	100.73	2675	2755	3.24	629-96-9
15	Oxirane, hexadecyl-	103.93	2836	-	0.51	7390-81-0
16	Cycloartenol acetate	107.96	3089	3389	1.98	1259-10-5
Total Identified (%)					100.00	

^1^ RT: Retention time, ^2^ RI: Retention indices as determined using a DB5-MS capillary column.

## Data Availability

Not applicable.

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
