# Peer review of "Chemical Composition of Salix koreensis Anderss Flower Absolute and Its Skin Wound Healing Activities In Vitro"

_plants, 2022, doi:10.3390/plants11030246_

Round 1

Reviewer 1 Report

Response to plants-1529083 Chemical Composition of Salix koreensis Anderss Flower 2 Absolute and Its Skin Wound Healing Activities In Vitro

Kim et al. investigated the effects of the flower absolute of SKA 15 (SKAFAb) on skin wound healing-associated responses in keratinocytes. The effects of SKAFAb on HaCaT cells were investigated using a Boyden chamber and 5-bromo-2’-deoxyuridine incorporation, sprouting, immunoblotting, enzyme-linked immunosorbent, and water-soluble tetrazolium salt assays. The results showed that SKAFAb promoted HaCaT cell proliferation, migration, and type I and IV collagen productions, increased sprout outgrowth and increased the phosphorylations of Akt, JNK, ERK1/2, and p38 MAPK in HaCaT cells. The study concluded that SKAFAb promoted keratinocyte proliferation and migration and induced collagen synthesis in keratinocytes.

The paper must be improved in many aspects and the authors should clarify some important issues, as follows:

  1. In the paper there are many abbreviations which should be explained. In the Introduction chapter the phrases are too long.
  2. Please explain why the authors used the immortalized keratinocytes for experiments and not normal keratinocytes.
  3. All verbs used to describe experiments and results must be in the past tense
  4. Please add in Introduction additional information about the effects of Salix koreensis Anderss Flower. What exactly means “flower absolute”?
  1. Please add information on the method of evaluation for cell migration and sprout formation and quantification of the results? The explanation is insufficient.
  2. The images of cells in Figure 3a are too small and difficult to see. Please add images larger and clearer images.
  1. The most accurate expression of western blot results is either by ratio between the size of the bands to β-actin or by expressing the ratio between the phosphorylated and the non-phosphorylated form.The quantification of non-phosphorylated forms is not otherwise justified. In addition, the western blot images do not contain β-actin blots.
  2. The “Discussion” chapter should be written more solid and present comparative the obtained results in this study with results of other studies which used the components from this extract. In present form the Discussion chapter it is a reproduction of the results.
  3. The paper contains a large number of grammar and syntax errors. Therefore, it has to be rewritten according with the scientific paper requirements.

Reviewer 2 Report

The manuscript titled “Chemical Composition of Salix koreensis Anderss Flower Absolute and Its Skin Wound Healing Activities In Vitro” is a comprehensive study focused on the potential application of SKA as a skin wound healing. In general, the manuscript is well redacted and organized.

Why do the authors dissolve the extract in ethanol rather than hexane? Do you obtain a homogeneous solution?

Why did the authors select hexane as extracting solvent?

As the extract is mostly composed of hydrophobic molecules, does the main achievements already expected?

Table 1: Despite characterizing the relative abundancy of each molecule, nowadays it is extremely important to quantify (at least) the most abundant ones. Can the authors perform the quantification of the main molecules?

Figures: please put “Salix koreensis” in italic form.

Line 177-186: Despite the bioactivity of nonanal, it should be noticed that authors are evaluating the effect of an extract. In fact, is the overall effect of it that is being evaluated. Therefore, in my opinion, this paragraph must be revised, highlighting this aspect.

As the extract is mostly composed of hydrophobic molecules,

Data Availability Statement: If the authors don’t have info to add, remove this section or add “Not applicable”.

Acknowledgements: If the authors don’t have info to add, remove this section or add “Not applicable”.

Author Response

Reviewer 1: Comments and Suggestions for Authors 

The manuscript titled “Chemical Composition of Salix koreensis Anderss Flower Absolute and Its Skin Wound Healing Activities In Vitro” is a comprehensive study focused on the potential application of SKA as a skin wound healing. In general, the manuscript is well redacted and organized. 

Comment(C) 1. Why do the authors dissolve the extract in ethanol rather than hexane? Do you obtain a homogeneous solution

(Response) Extracts of aromatic plants extracted with hydrocarbon type solvent such as hexane or pentane contain some unwanted waxy components (so-called ‘concrete’) and fatty esters. Concrete is concentrated form of hydrophobic and volatile (aromatic) components. However, the concrete is of poor quality due to the presence of wax components, which causes problems such as clouding because dissolution in fragrance formulations is limited. Therefore, the concrete has to be converted into a wax free and alcohol soluble volatile concentrate, known as ‘absolute’. Thus, as described in the Methods section of the present manuscript, the dark yellow waxy residue (concrete) obtained from Salix koreensis Anderss flowers was mixed with ethanol and left at -20°C for 12 h, filtered through a sintered funnel, and then the ethanol was removed by evaporation at 35°C to leave a light-yellow anhydrous wax (SKAFAb). which was stored at -80°C for experiments. In addition, we used the absolute dissolved in DMSO for each test. 

[Reference]

1. Rout PK, Naik S, Rao YR. Liquid CO2 extraction of flowers and fractionation of floral concrete of Michelia champaca Linn. J. SUPERCRIT. FLUID. 2011, 56(3), 249-252.

2. Prakash O, Sahoo D, Rout PK. Liquid CO2 extraction of Jasminum grandiflorum and comparison with conventional processes. Nat Prod Commun. 2012, 7(1), 89-92.

C2Why did the authors select hexane as extracting solvent? 

(Response) Volatile (aromatic)-related compounds such as terpenes and sesquiterpenes, but also fatty acids and their methyl esters, paraffins and other high molecular weight compounds are extracted from aromatic plant materials using volatile solvents such as n-hexane, toluene, benzene and petroleum ether. Among various non-polar solvents used for extraction of volatile compounds, n-hexane has relatively low toxicity, can effectively volatile components, and is easy to be removed. Thus, in the present study, we extracted ‘absolute’ extracted from Salix koreensis Anderss flowers using n-hexane. 

C3As the extract is mostly composed of hydrophobic molecules, does the main achievements already expected? 

(Response) Yes, we did. We extracted ‘absolute’ extracted from Salix koreensis Anderss flowers using non-polar solvent n-hexane. Considering n-hexane as mentioned above, it was expected that most of the components to be isolated from the absolute extract would be molecules with affinity for non-polar solvents. In addition, based on this research experience of our team for 10, we tried to extract non-polar, volatile, and aromatic components. Thus, our research team conducted the present study to investigate the skin wound healing activity by absolute (containing components of volatile hydrophobic molecules) extracted from the flowers of Salix koreensis Anderss among native plants in Korea.   

C4Table 1: Despite characterizing the relative abundancy of each molecule, nowadays it is extremely important to quantify (at least) the most abundant ones. Can the authors perform the quantification of the main molecules? 

(Response) In the present study, the results of component analysis of Salix koreensis Anderss flower absolute are shown as relative contents (%). As the reviewer commented, quantification of each component is very important point. Further study will need to quantify the identified absolute components by comparing them with components that were commercially sold or by more accurately isolating them for components that are not sold. In addition, the stability and activities of the quantified absolute components will also need to be clarified. Based on our research experience, we believe that each component can be quantified. We thank the reviewer for the valuable and constructive comments.

C5Figures: please put “Salix koreensis” in italic form.

(Response) Thanks. We did that. 

C6Line 177-186: Despite the bioactivity of nonanal, it should be noticed that authors are evaluating the effect of an extract. In fact, is the overall effect of it that is being evaluated. Therefore, in my opinion, this paragraph must be revised, highlighting this aspect. 

(Response) We revised it according to the reviewer’s comment (Line 215-226 in revised manuscript).

C7As the extract is mostly composed of hydrophobic molecules,

C7-1. Data Availability Statement: If the authors don’t have info to add, remove this section or add “Not applicable”

(Response) Thanks. We added the section. 

C7-2. Data Availability Statement: If the authors don’t have info to add, remove this section or add “Not applicable”

(Response) Thanks. We did not add the section when we initially submitted the manuscript. 

Reviewer 3 Report

The manuscript is overall well-written and the results support the conclusion.

However, there are minor corrections to carry out for improving manuscript quality.

-The aim description has to be improved in both abstract and introduction section.

-The plant name has to be checked and corrected. In different parts of the manuscript the plant name is not reported in italics.

-Additionally, authors should also check the use of acronyms, with the attention to mention them in extrenso at the first appearance in the text.

-Regarding the statistical analysis, authors mentioned the use of ANOVA followed by post hoc test (Tukey test); however, in the figure captions they reported only the P values related to post hoc test and not the ANOVA P value.

-The conclusion section should be moved as separate paragraph at the end of the manuscript (after paragraph 4.11).

Author Response

Reviewer 2Comments and Suggestions for Authors 

The manuscript is overall well-written and the results support the conclusion. However, there are minor corrections to carry out for improving manuscript quality. 

Comment (C) 1. The aim description has to be improved in both abstract and introduction section. 

(Response) We revised it according to the reviewer’s comment.

C2. The plant name has to be checked and corrected. In different parts of the manuscript the plant name is not reported in italics.  

(Response) Thanks. We revised those. 

C3. Additionally, authors should also check the use of acronyms, with the attention to mention them in extrenso at the first appearance in the text.  

(Response) Thanks. We checked and revised those. 

C4. Regarding the statistical analysis, authors mentioned the use of ANOVA followed by post hoc test (Tukey test); however, in the figure captions they reported only the P values related to post hoc test and not the ANOVA P value. 

(Response) We revised those in each caption according to the reviewer’s comments. 

C5. The conclusion section should be moved as separate paragraph at the end of the manuscript (after paragraph 4.11).

(Response) We revised the manuscript according to the reviewer’s comment.

Round 2

Reviewer 1 Report

The authors have satisfactory answered to all questions mentioned.

Author Response

January 13, 2022

Plants

Editorial Office:

Dear Sir or Madam:

Thank you for your letter that was sent to me along with the reviewer’s comments on our revised manuscript (Manuscript ID: plants-1529083).

             I send you the revised manuscript. Based on the valuable comments from reviewer, we did our best to revise the manuscript. Our responses to the reviewer’s comments is summarized in the separated sheets. In the revised manuscript, revisions were highlighted using the "Track Changes" function of Microsoft Word.      

             The revised manuscript, or part of it, has neither been published nor is currently under consideration for publication by any other journal. All authors have read the revised manuscript and agreed with the contents of the revised manuscript. They also approved its returning to Plants. None of the authors have any conflict of interest.

             I and all other authors hope that the revised manuscript is now suitable for publication of Plants and are very grateful to you and reviewers.

             I am looking forward to getting a positive reply from you soon.

Sincerely yours,

Hwan Myung Lee, Ph.D., Professor

Division of Cosmetic and Biotechnology

College of Life and Health Sciences

Hoseo University, Asan 31499, Republic of Korea

Responses to reviewer’ comments (2nd Revision)  

Manuscript No: plants-1529083

Title: Chemical composition of Salix koreensis Anderss flower absolute and its skin wound healing activities in vitro

Authors: Nan Young Kim et al.

Reviewer 2: Comments and Suggestions for Authors

Comment(C) 1. The authors addressed some of the suggestions. It is obvious that using hydrophobic solvents, hydrophobic molecules will be extracted. Also, the authors should avoid mentioning this study under the scope of aromas/fragrances as only Verbenone was identified. Furthermore, the selected methodology is not for the extraction of aromas/fragrances (e.g. hydrodistillation, steam distillation, supercritical CO2 among other alternative/green designer solvents).

(Response) Aroma (volatile) compounds can be extracted by various extraction methods including steam distillation, hydrodistillation, supercritical CO2 extraction, cold pressing, and solvent extraction (2.3). The solvent extract is widely known as a representative aroma (volatile) compound extraction method that has been used for a long time (2,3,6).

             Extraction from aromatic plants using volatile solvents such as n-hexane, toluene, benzene and petroleum ether can produce volatile (aromatic)-related compounds such as terpenes and sesquiterpenes as well as also fatty acids and other high molecular weight compounds. Extraction from aromatic plants using nucleic acid or pentane produces a waxy extract called 'concrete'. Concrete is a concentrated form of hydrophobic and volatile (aromatic) components isolated in a hydrocarbon type solvent such as hexane or pentane. In this extraction process, some unwanted waxy components and fatty esters are also co-extracted. The concrete obtained is of lower quality due to the presence of the waxy components which cause problems such as clouding during storage in fragrance formulations resulting from their limited solubility. Therefore, the concrete has to be converted into a wax free and alcohol soluble volatile (aromatic) concentrate, known as ‘absolute’ (1,4,6) as in the Methods of the present study.

             Steam distillation or hydrodistillation extraction can lead to the decomposition of thermally labile compounds in plant materials due to the time consuming process performed at high temperatures. Therefore, it is known that extraction methods can be imperfect for the extraction of essential oils (5). However, common solvents such as hexane, methanol, and ethanol can be used to extract fragile and delicate plant flower materials that cannot be extracted with heat or steam (2). Moreover, essential oils have hydrophobic properties and lower density than water, so they are lipophilic and soluble in organic solvents, and are immiscible with water (2).

             Thus, in the present study, we extracted Salix koreensis Anderss flower absolute (SKAFAb) and investigated biological activities of the absolute in skin. Moreover, based on Adams’s analysis data (7) and other essential oil-related research articles we compared the retention index values of 16 identified components from SKAFAb to reconfirm that they are aromatic substances. 

[References]

  1. Rout PK, Naik S, Rao YR. Liquid CO2 extraction of flowers and fractionation of floral concrete of Michelia champaca Linn. J. SUPERCRIT. FLUID. 2011, 56(3), 249-252
  2. Aziz ZAA, Ahmad A, Setapar SHM, Karakucuk A, Azim MM, Lokhat D, Rafatullah M, Ganash M, Kamal MA, Ashraf GM. Essential Oil: Extraction Techniques, Pharmaceutical and Therapeutic Potential - A Review. Curr Drug Metab. 2018, 19(13), 1100-1110.
  3. El Asbahani A, Miladi K, Badri W, Sala M, Aït Addi EH, Casabianca H, El Mousadik A, Hartmann D, Jilale A, Renaud FN, Elaissari A.Essential oils: from extraction to encapsulation.Int J Pharm. 2015, 483(1-2), 220-243.
  4. Baydar H, Kineci S, Scent Composition of Essential Oil, Concrete, Absolute and Hydrosol from Lavandin (Lavandula x intermedia Emeric ex Loisel.). J. Essent. Oil-Bear. Plants. 2008, 12(2), 131-136
  5. Danh LT, Han LN, Anh Triet ND, Zhao J, Mammucari R, Foster N. Comparison of Chemical Composition, Antioxidant and Antimicrobial Activity of Lavender (Lavandula angustifolia L.) Essential Oils Extracted by Supercritical CO2, Hexane and Hydrodistillation. Food and bioprocess, 2013. 1-9.
  6. Rout PK, Misra R, Sahoo S, Sree A, Rao YR. Extraction of kewda (Pandanus fascicularis Lam.) flowers with hexane: composition of concrete, absolute and wax. Flavour Fragr. J. 2005, 20, 442–444.
  7. Adams, RP. Identification of essential oil components by gas chromatography/mass spectrometry. 4th ed.; Allured Publishing Cor-poration: Carol Stream, Illinois, USA, 2007).

             In addition, in the present study, 16 components including verbenone isolated from SKAFAb were also identified in essential oils of other plant species extracted by steam distillation or hydrodistillation methods as shown in Table 1 below. This suggest that components of aromas/fragrances may contain various aldehydes, ketones, and alcohol-based components as well as terpenes, although we described our study under the scope of aromas/fragrances in the presented manuscript.

Table 1

No

Component name

CAS No.

Reference No.

1

Nonanal

124-19-6

3

aldehyde

2

Verbenone

1196-01-6

3

mono terpene

3

Hexadecanal

629-80-1

4

aldehyde

4

Tetradecanal

124-25-4

3

Aldehyde

5

2-Nonadecanone

629-66-3

5

ketone

6

Pentadecanal

2765-11-9

3

aldehyde

7

Eicosane

112-95-8

1

alkane 

8

Tricosane

638-67-5

2

alkane

9

Docosane

629-97-0

2

alkane

10

9-Tricosene, (Z)-

27519-02-4

3

11

Pentacosane

629-99-2

2

alkane

12

Tetracosane

646-31-1

2

alkane

13

Tetracosane, 11-decyl-

(other name: 11-n-Decyltetracosane)

55429-84-0

9

14

1-Eicosanol

629-96-9

6

alcohol 

15

Oxirane, hexadecyl-

7390-81-0

7

16

Cycloartenol acetate

1259-10-5

8

[References]

  1. Baydar H, Kineci S. Scent composition of essential oil, concrete, absolute and hydrosol from Lavandin (Lavandula x intermedia Emeric ex Loisel.). J. Essent. Oil-Bear. Plants. 2008, 12, 131-136
  2. Tsasi G, Mailis T, Daskalaki A, Sakadani E, Razis P, Samaras Y, Skaltsa H. The effect of harvesting on the composition of essential oils from five varieties of Ocimum basilicum L. cultivated in the Island of Kefalonia, Greece. Plants (Basel). 2017, 6, 41. 
  3. Saroglou V, Dorizas N, Kypriotakis Z, Skaltsa HD. Analysis of the essential oil composition of eight Anthemis species from Greece. J Chromatogr A. 2006, 1104, 313-322. 
  4. Ferhat MA, Tigrine-Kordjani N, ChematS, Meklati BY, Chemat F. Rapid Extraction of Volatile Compounds Using a New Simultaneous Microwave Distillation: Solvent Extraction Device. Chromatographia. 2007, 217–222
  5. Kim JS, Chung HY. GC-MS analysis of the volatile components in dried boxthorn (Lycium chinensis) fruit. Journal of the Korean Society for Applied Biological. 2009, 52, 516–524.
  6. Nogueira PC, Bittrich V, Shepherd GJ, Lopes AV, Marsaioli AJ. The ecological and taxonomic importance of flower volatiles of Clusia species (Guttiferae). Phytochemistry. 2001, 56, 443-452.
  7. Oxygenated chemical constituents, brine shrimp toxicity and free radical scavenging activity of leaf essential oils of Alchornea cordifolia (Schumach. & Thonn.) Mull. Arg. G. K. Oloyede *, P. A. Onocha International Journal of Essential Oil Therapeutics (2010) 4, 104-107.
  8. Ante I, Aboaba S, Siddiqui H, Iqbal Choudhary M. Essential Oils of the Leaf, Stem-Bark, and Nut of Artocarpus camansi: Gas chromatography-mass spectrometry analysis and activities against multidrug-resistant bacteria. Journal of Herbs, Spices & Medicinal Plants. 2016, 22, 203-210.
  9. Zhang Y, Wang Z. Influence of drying methods on chemical composition of the essential oil of Glechoma longituba. Chemistry of Natural compounds. 2007, 43. 625-628.

C2. As it was the first time that those properties of Salix koreensis flower extract were reported, the chemical composition (quantitative) seems to be imperative at a Q1 journal such as Plants.

(Response) As the reviewer suggested, we analyzed the quantitative chemical composition of each component of SKAFAb. Based on the GC/MS analysis result, 1) the peak area ratio of each component identified was calculated, and 2) these were converted into mg/g as followings: 1) Proportion of each component (% ratio) = Peak area of each component/ Total peak area x 100; 2) Amount of each component contained in 1g of SKAFAb = % ratio of each component/100 X 1000 mg (1 g, SKAFAb). The analysis results is showed in Table 2 below

Table 2.

No.

Component name

RTa

RIb

Cas No.

Area (%)

Amount (mg/g total )

Observed

Literature

1

Nonanal

22.24

1104

1104

124-19-6

1.29

12.9

2

Verbenone

28.94

1201

1204

1196-01-6

3.07

30.7

3

Hexadecanal

65.69

1814

1814

629-80-1

1.25

12.5

4

Tetradecanal

75.65

2018

1940

124-25-4

0.63

 6.3

5

2-Nonadecanone

79.45

2101

2101

629-66-3

7.13

71.3

6

Pentadecanal-

84.76

2222

-

2765-11-9

0.39

 3.9

7

Eicosane

86.52

2263

2000

112-95-8

3.02

30.2

8

Tricosane

88.08

2300

2300

638-67-5

10.00

100.0

9

Docosane

94.64

2465

2200

629-97-0

16.39

163.9

10

9-Tricosene, (Z)-

94.97

2474

2272

27519-02-4

18.72

187.2

11

Pentacosane

95.99

2500

2500

629-99-2

2.67

 26.7

12

Tetracosane

98.19

2574

2400

646-31-1

1.19

 11.9

13

Tetracosane, 11-decyl-;

100.50

2665

-

55429-84-0

0.38

  3.8

14

1-Eicosanol

100.73

2675

2755

629-96-9

2.27

 22.7

15

Oxirane, hexadecyl-

103.93

2836

-

7390-81-0

0.36

  3.6

16

Cycloartenol acetate

107.96

3089

3389

1259-10-5

1.39

 13.9

Identified

70.15

701.5

Unidentified

29.85

288.5

Total

100.00

1000.0

RTa: Retention time; Rib: Retention indices determined using a DB5-MS capillary columm

C3. Additionally, the chemical composition is poorly linked/discussed with their potential as wound healing agents. In this regard, at the present form, this analysis seems to be an "island" inside a study regarding the biological activities of an extract?

 (Response) We revised the Discussion section according to the reviewer’s comments.

C4. Table 1: Can the authors add the quantification of the found molecules and improve their discussion in this regard (I mean, in this manuscript)?

(Response) As the reviewer commented, we analyzed and showed the quantification of the found molecules as in our response (C2) described above and also improved the Discussion section (Page 7/13, 1st and 2nd paragraph)

Reviewer 2 Report

The authors addressed some of the suggestions. It is obvious that using hydrophobic solvents, hydrophobic molecules will be extracted. Also, the authors should avoid mentioning this study under the scope of aromas/fragrances as only Verbenone was identified. Furthermore, the selected methodology is not for the extraction of aromas/fragrances (e.g. hydrodistillation, steam distillation, supercritical CO2 among other alternative/green designer solvents).

As it was the first time that those properties of Salix koreensis flower extract were reported, the chemical composition (quantitative) seems to be imperative at a Q1 journal such as Plants.

Additionally, the chemical composition is poorly linked/discussed with their potential as wound healing agents. In this regard, at the present form, this analysis seems to be an "island" inside a study regarding the biological activities of an extract.

Can the authors add the quantification of the found molecules and improve their discussion in this regard (I mean, in this manuscript)?

Author Response

(The authors gave the same response as above.)

Round 3

Reviewer 2 Report

It would be more productive if the authors kindly revise the concept of "aromatic" and what an aromatic compound truly is.

n-hexane has the capacity of extracting aromatic molecules (among others), yes. But when the authors evaporate the solvent they will be lost, as perfectly summarized in table 2 (and described in M&M section).

Table 2 also reports absolute values instead of mean and SD values, which is crucial to believe that replicates were analysed.

When extracting aromas from plant materials, extracts are mostly composed of aromatic compounds and lower amounts of volatile oils (such as LMW alkanes). Taking advantage of literature without respecting its core of knowledge is not science.